

# Musculoskeletal models of a human and bonobo finger: parameter identification and comparison to in vitro experiments

Alexander Synek[1], Szu-Ching Lu[2,3], Evie E. Vereecke[4], Sandra Nauwelaerts[5,8], Tracy L. Kivell[3,6] and Dieter H. Pahr[1,7]

[1] Institute of Lightweight Design and Structural Biomechanics, TU Wien, Vienna, Austria
[2] Laboratory for Innovation in Autism, School of Education, University of Strathclyde, Glasgow, United Kingdom
[3] Animal Postcranial Evolution Lab, Skeletal Biology Research Centre, School of Anthropology and Conservation, University of Kent, Canterbury, United Kingdom
[4] Department of Development and Regeneration, University of Leuven, Kortrijk, Belgium
[5] Department of Biology, University of Antwerp, Wilrijk, Belgium
[6] Department of Human Evolution, Max Planck Institute for Evolutionary Anthropology, Leipzig, Germany
[7] Department of Anatomy and Biomechanics, Karl Landsteiner Private University of Health Sciences, Krems an der Donau, Austria
[8] Center for Research and Conservation KMDA, Astridplein, Antwerpen, Belgium

Corresponding author
Alexander Synek,
asynek@ilsb.tuwien.ac.at

## ABSTRACT

**Introduction**. Knowledge of internal finger loading during human and non-human primate activities such as tool use or knuckle-walking has become increasingly important to reconstruct the behaviour of fossil hominins based on bone morphology. Musculoskeletal models have proven useful for predicting these internal loads during human activities, but load predictions for non-human primate activities are missing due to a lack of suitable finger models. The main goal of this study was to implement both a human and a representative non-human primate finger model to facilitate comparative studies on metacarpal bone loading. To ensure that the model predictions are sufficiently accurate, the specific goals were: (1) to identify species-specific model parameters based on in vitro measured fingertip forces resulting from single tendon loading and (2) to evaluate the model accuracy of predicted fingertip forces and net metacarpal bone loading in a different loading scenario.

**Materials & Methods**. Three human and one bonobo (*Pan paniscus*) fingers were tested in vitro using a previously developed experimental setup. The cadaveric fingers were positioned in four static postures and load was applied by attaching weights to the tendons of the finger muscles. For parameter identification, fingertip forces were measured by loading each tendon individually in each posture. For the evaluation of model accuracy, the extrinsic flexor muscles were loaded simultaneously and both the fingertip force and net metacarpal bone force were measured. The finger models were implemented using custom Python scripts. Initial parameters were taken from literature for the human model and own dissection data for the bonobo model. Optimized model parameters were identified by minimizing the error between predicted and experimentally measured fingertip forces. Fingertip forces and net metacarpal bone loading in the combined loading scenario were predicted using the optimized models and the remaining error with respect to the experimental data was evaluated.

**Results**. The parameter identification procedure led to minor model adjustments but considerably reduced the error in the predicted fingertip forces (root mean square error reduced from 0.53/0.69 N to 0.11/0.20 N for the human/bonobo model). Both models remained physiologically plausible after the parameter identification. In the combined loading scenario, fingertip and net metacarpal forces were predicted with average directional errors below 6° and magnitude errors below 12%.

**Conclusions**. This study presents the first attempt to implement both a human and non-human primate finger model for comparative palaeoanthropological studies. The good agreement between predicted and experimental forces involving the action of extrinsic flexors—which are most relevant for forceful grasping—shows that the models are likely sufficiently accurate for comparisons of internal loads occurring during human and non-human primate manual activities.

# INTRODUCTION

Knowledge of internal loads of the finger such as joint loads or muscle forces are highly important in many scientific disciplines. For instance, large joint loads provide evidence for joints at particular risk to develop osteoarthritis (*Goislard de Monsabert et al., 2014*); muscle forces acting during different grasps help to understand the etiology of common injuries, such as pulley rupture during rock climbing (*Vigouroux et al., 2008*; *Roloff et al., 2006*), and to design more ergonomic products (*Ikeda, Kurita & Ogasawara, 2009*; *Vigouroux, Domalain & Berton, 2011*). Recently, internal loading of the finger has also become increasingly relevant to interpret fossil remains of human ancestors, or hominins (*Rolian, Lieberman & Zermeno, 2011*). Since bone adapts to mechanical loading (*Huiskes, 2000*; *Wolff, 2010*; *Frost, 1987*), knowledge of the loading conditions might reveal valuable information about the behaviour of extinct species. While activity-related differences of finger bone morphology between primate species has been extensively investigated in the past (*Susman, 1979*; *Hunt, 1991*; *Tsegai et al., 2013*; *Chirchir et al., 2017*), there is still a lack of data on the actual bone loads during activities most relevant to interpreting extant and extinct ape morphology such as climbing, knuckle-walking, and tool use.

Assessment of internal finger loading during these activities poses particular challenges that exceed those of most medical and ergonomic studies. Typically, musculoskeletal models are used in these studies to compute joint loads and muscle forces at the finger as both are ethically and logistically challenging to measure in vivo (*Goislard de Monsabert et al., 2014*; *Vigouroux et al., 2008*). Most previous musculoskeletal models are based on human anatomy and model parameters can be obtained from cadaveric studies (*An et al., 1979*; *Chao et al., 1989*). However, accurate predictions of internal loads during non-human primate activities would require additional finger models specific to non-human primate anatomy. For instance, differences in the ratio of bone segment lengths across primates might have a considerable effect on finger biomechanics (*Feix et al., 2015*; *Susman, 1979*).
Furthermore, there are substantial differences in hand musculature between humans and other primates, including variation in where muscles attach, their architecture and even absence/presence of particular muscles (*Marzke et al., 1999*; *Van Leeuwen et al., 2018*; *Lemelin & Diogo, 2016*). If not accounted for, all of these anatomical differences might lead to inaccurate model predictions. Still, the authors are not aware of a complete non-human primate finger model that allows predicting joint loads or muscle forces. *Schaffelhofer et al. (2015)* modified a previously presented upper extremity model (*Holzbaur, Murray & Delp, 2005*) to match the macaque anatomy, but intrinsic hand muscles were not included and the model adaptations were limited to the finger segment lengths.

In addition to the requirement of models accurately representing both human and non-human primate anatomy, model validation using experimental data is crucial to ensure realistic predictions (*Hicks et al., 2014*). Previous efforts of human finger model validation aimed at different output quantities and generally found a good agreement between experiments and model predictions: *Kociolek & Keir (2011)* and *Lee et al. (2014)* compared predicted and in vitro measured moment arms of tendons at individual joints, *Qiu & Kamper (2014)* showed that predicted measured fingertip forces agree well with those measured in vitro, and electromyographic measurements demonstrated the validity of predicted muscle activation patterns (*Valero-Cuevas, Zajac & Burgar, 1998*; *Ikeda, Kurita & Ogasawara, 2009*). However, a direct validation of joint or net bone loading is currently missing, even though these quantities are of particular relevance to robust functional interpretations of bony morphology.

The main goal of this study was to fill these gaps by implementing human and bonobo musculoskeletal finger models that enable comparative studies on internal finger loading of humans and non-human primates. Bonobos (*Pan paniscus*) were selected as they are genetically highly similar to humans (*Prüfer et al., 2012*) but still engage in locomotor behaviours such as climbing, suspension, and knuckle-walking (*Doran, 1996*) that are relevant to reconstructing behaviour in fossil hominins. To ensure the models provide valid results, the specific study goals were to: (1) identify model parameters that minimize the error between predicted and in vitro measured fingertip forces and (2) to compare fingertip and metacarpal bone load predictions of the adjusted models to experimental measurements in different load cases for validation. Additionally, the human and bonobo model shall be compared with each other to investigate whether or not the use of a bonobo specific model is warranted.

## MATERIALS & METHODS

### Study outline

Parameter identification was performed by adjusting the parameters of a human and bonobo third digit model to best match fingertip forces measured in vitro in four postures while loading each muscle/tendon individually (Fig. 1A). In a second step, multiple tendons were loaded simultaneously in the same four postures and the measured fingertip forces and net metacarpal bone loading (i.e., the total force acting on the metacarpal bone) were compared to the model predictions (Fig. 1B). Finally, differences between the human and bonobo models were evaluated in these combined tendon loading conditions.

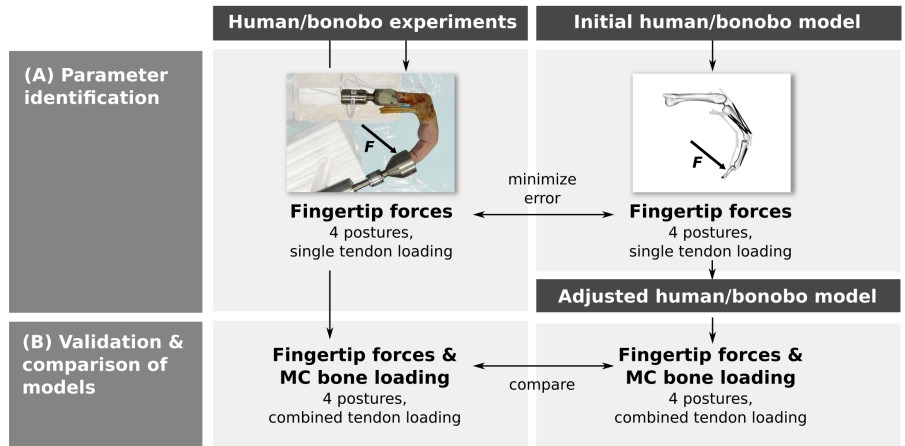

**Figure 1** **Outline of the study.** Parameters of a human and bonobo finger model were first identified by adjustment to in vitro experimental data (A) and then validated in different load cases (B). Additionally, human and bonobo models were compared to identify species-related differences. MC: metacarpal.

## In vitro experiments

The fingertip and net metacarpal bone forces were assessed using a previously developed custom experimental setup (*Lu et al., 2018*) (see Fig. 2). This setup permits the mounting of a dissected cadaveric finger both at the metacarpal bone and fingertip to ensure a fixed, static posture and to apply load by attaching weights to individual tendons. Fingertip forces and net metacarpal bone forces were measured using a six-axis load cell (Nano 17-E, ATI Industrial Automation, Apex, NC, USA) which was positioned either at the proximal or distal bone clamp (labelled "Load cell location 1" and "Load cell location 2", respectively, in Fig. 2).

### Study sample

The study sample comprised three third digits of fresh frozen cadaveric human hand specimens (age: 89.7 ± 4.0 years; gender: two female, one male; side: left) and one third digit of a fresh frozen bonobo hand specimen (taxon: *Pan paniscus*; age: 8 years; gender: female; side: left). Human samples were obtained via the Human Body Donation Programme of the University of Leuven, Belgium and the bonobo sample was made available by the Antwerp Zoo by Centre for Research and Conservation, Royal Zoological Society Antwerp (KMDA/RZSA) as part of the Bonobo Morphology Initiative 2016. The Bonobo Morphology Initiative made the cadavers of bonobos euthanized for medical reasons available for research.

### Specimen preparation

The digits were disarticulated from the hands at the carpometacarpal joints and soft tissues were removed to identify the tendons of all intrinsic and extrinsic muscles as listed in Table 1. The soft tissues around the metacarpophalangeal (MCP) joint were kept intact to the maximum extent possible to maintain physiological conditions. Sutures were applied to each tendon using the Clove-Hitch technique (*Abraham et al., 2012*). In cases

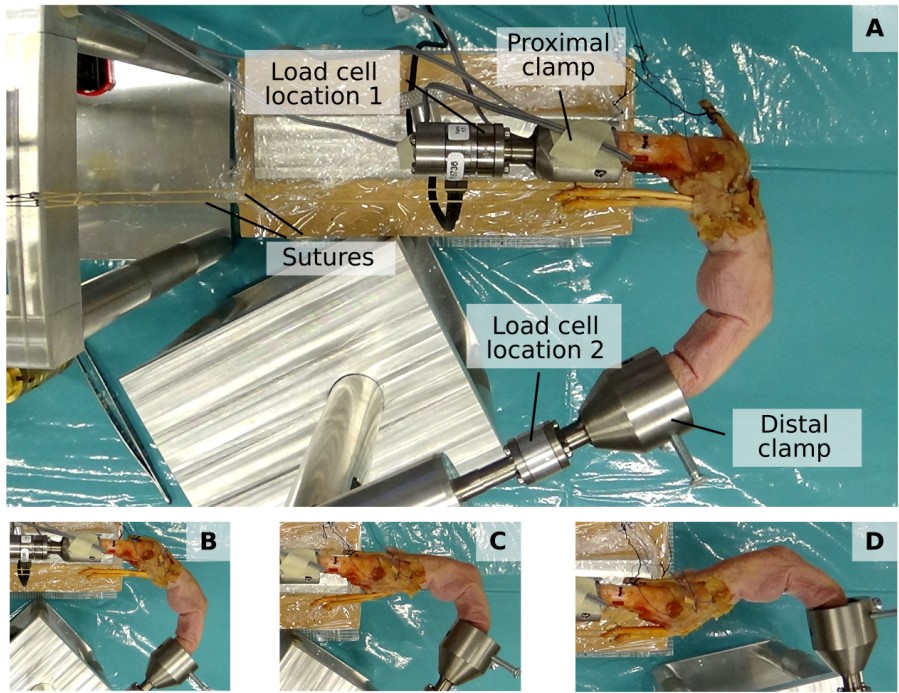

**Figure 2 Experimental setup.** The proximal and distal clamp fixate a dissected cadaveric finger in a static posture. Individiual tendons can be loaded by applying weights to attached sutures. A load cell enables measuring both fingertip forces when positioned at "Load cell location 2" and net metacarpal bone loading when positioned at "Load cell location 1". (A to D) show the four different postures, namely major flexion (A), minor flexion (B), hook (C), and hyperextension (D).

where intrinsic muscle tendons clearly split into two parts inserting either at the extensor mechanism (EM) or proximal phalanx (PP) base, sutures were applied to each part of the tendon (see Table 1).

### Experimental design

Each specimen was mounted to the experimental setup and placed in four postures that vary in their degree of flexion/extension and simulate common postures used by both humans and African apes (see Fig. 2 and Table 2): (1) major flexion, (2) minor flexion, (3) hook grip, and (4) hyperextension. Respective joint angles were set using a goniometer. In each posture, the tendons were loaded in proportion to the maximum muscle force $t_{max}$, as estimated from the muscle specific PCSA and the maximum specific muscle tension of 45 N/cm$^2$ (*Holzbaur, Murray & Delp, 2005*). Human muscle PCSA data were taken from *Chao et al. (1989)* and the bonobo muscle PCSAs were obtained from a dissection study on the contralateral arm of the same bonobo cadaver (Article S1). The sutures were aligned in parallel to the long axis of the metacarpal bone (see Fig. 2) to best approximate physiological loading conditions.

For parameter identification, the load cell was mounted at the fingertip clamp and forces were recorded while each individual tendon was loaded to 5% of the maximum muscle force (see Table 1 for muscle-specific weights). In the combined tendon loading

**Table 1  Muscles and tendons loaded in the in vitro experiments.** Intrinsics with split tendons inserting into either the extensor mechanism (EM) or the proximal phalanx base (PP) are labelled accordingly. Loads were applied in proportion to the PCSA as taken from *Chao et al. (1989)* for the human fingers and own dissection data for the bonobo finger (see Article S1). Note that values for the UI (PP) of the human specimens are omitted as the UI did not insert into the proximal phalanx.

| Species | Muscle/tendon | PCSA (cm$^2$) | Mass (g) 5% $t_{max}$ | 2% $t_{max}$ |
|---|---|---|---|---|
| Bonobo | FDS | 3.5 | 800.0 | 300.0 |
| | FDP | 2.9 | 650.0 | 300.0 |
| | EDC | 1.1 | 250.0 | – |
| | LU | 0.2 | 40.0 | – |
| | RI (EM) | 0.8 | 200.0 | – |
| | RI (PP) | 1.5 | 350.0 | – |
| | UI (EM) | 0.8 | 200.0 | – |
| | UI (PP) | 0.9 | 200.0 | – |
| Human | FDS | 4.2 | 950.0 | 300.0 |
| | FDP | 4.1 | 950.0 | 300.0 |
| | EDC | 1.7 | 400.0 | – |
| | LU | 0.2 | 45.0 | – |
| | RI (EM) | 1.4 | 325.0 | – |
| | RI (PP) | 1.4 | 325.0 | – |
| | UI (EM) | 2.2 | 500.0 | – |
| | UI (PP) | – | – | – |

Notes.
EDC, extensor digitorum communis; FDP, flexor digitorum profundus; FDS, flexor digitorum superficialis; RI, radial interosseus; UI, ulnar interosseus; LU, lumbrical; PCSA, physiological cross sectional area.

**Table 2  Postures used in this study including joint angles.** Ulnar/radial deviation was 0° in all postures. Joint angles were the same for the bonobo and human specimens except for major flexion (marked with *). Major flexion joint angles were modified for the human fingers due to the specimen range of motion, such that DIP/PIP/MCP angles were set to 25°/57°/55°, respectively.

| Posture | DIP flexion (°) | PIP flexion (°) | MCP flexion (°) |
|---|---|---|---|
| Major flexion* | 40.0 | 50.0 | 60.0 |
| Minor flexion | 35.0 | 45.0 | 40.0 |
| Hook grip | 50.0 | 65.0 | 0.0 |
| Hyperextension | 45.0 | 50.0 | −20.0 |

Notes.
DIP, distal interphalangeal; PIP, proximal interphalangeal; MCP, metacarpophalangeal.

scenario, the fingertip forces and net metacarpal bone loads were recorded while the flexor digitorum profundus (FDP) and flexor digitorum superficialis (FDS) muscle were loaded simultaneously at two different force levels, namely 5% and 2% of the maximum muscle force (see Table 1). The tendons were loaded with only 2% to 5% of the maximum muscle force to prevent ruptures at the sutures.

### Data acquisition and processing

A compact data acquisition system (NI cDAQ-9174, National Instruments, Austin, TX, USA) and a custom LabVIEW (National Instruments, Austin, TX, USA) program were
used to measure forces both in the loaded and unloaded finger. The fingertip and net metacarpal bone forces were then computed as the difference of the measurements in the loaded and unloaded state to exclude gravitational forces and constraint forces resulting from specimen positioning.

After the experiments were conducted, the intended tendon load (governed by the attached weights) was compared to the true tendon loading computed based on static equilibrium equations using the fingertip force and net metacarpal bone loading available from the combined tendon loading scenarios. Deviations between intended and computed values were found and larger than expected. These deviations were attributed to friction at a pulley in the experimental setup which deflects the suture as required to apply the weights. In order to diminish the resulting error, a linear correction factor $c$ was calculated:

$$c = \frac{1}{n} \sum_{i=1}^{n} \left( \frac{\|F_{\text{tip},i} + F_{\text{bone},i}\|}{m_i \cdot g} \right) \tag{1}$$

In the above equation, $m_i \cdot g$ is the weight attached to the tendons during load case $i$ computed from mass $m_i$ and the gravitational constant $g = 9.81 \text{ m/s}^2$, $F_{\text{tip},i}$ is the respective fingertip force, $F_{\text{bone},i}$ is the net metacarpal bone loading, and $\|\cdot\|$ is the Euclidean norm. Including all four specimens, all four postures, and both load levels (i.e., $n = 32$) led to a correction factor of 0.835. This factor was used to correct all tendon tensions for both the combined and single tendon load cases.

## Musculoskeletal finger models
### Kinematics

Both the human and bonobo musculoskeletal models were generated based on the kinematic description and tendon via points provided by *An et al. (1979)* and implemented using custom Python scripts. The kinematics comprise of three movable (proximal, middle, and distal phalanx) and one fixed (metacarpal) bone segments interconnected by three joints, namely the MCP, the proximal interphalangeal (PIP) and the distal interphalangeal (DIP) joint (see Fig. 3). PIP and DIP joints were modelled as hinge joints with one degree of freedom (flexion/extension) and the MCP joint as a condylar joint with two rotational degrees of freedom (flexion-extension and radial/ulnar deviation). All flexion/extension joint axes were fixed and parallel to each other and the two MCP joint axes were intersecting and perpendicular.

### Muscles and tendons

Six muscles actuate the finger models (see Fig. 3): the three extrinsic muscles FDP, FDS and extensor digitorum communis (EDC), and the three intrinsic muscles radial interosseus (RI), ulnar interosseus (UI) and lumbrical (LU). The extensor mechanism was included using the common Winslow's rhombus simplification (*Zancolli, 1979*; *Valero-Cuevas, Zajac & Burgar, 1998*; *Synek & Pahr, 2016*). It consists of two slips and two bands, namely the central slip, terminal slip, ulnar band, and radial band (see Fig. 3). The tendon paths were approximated by straight line segments using via points proximal and distal to each joint as described by *An et al. (1979)* (see also Fig. 4). The coordinates of the proximal and

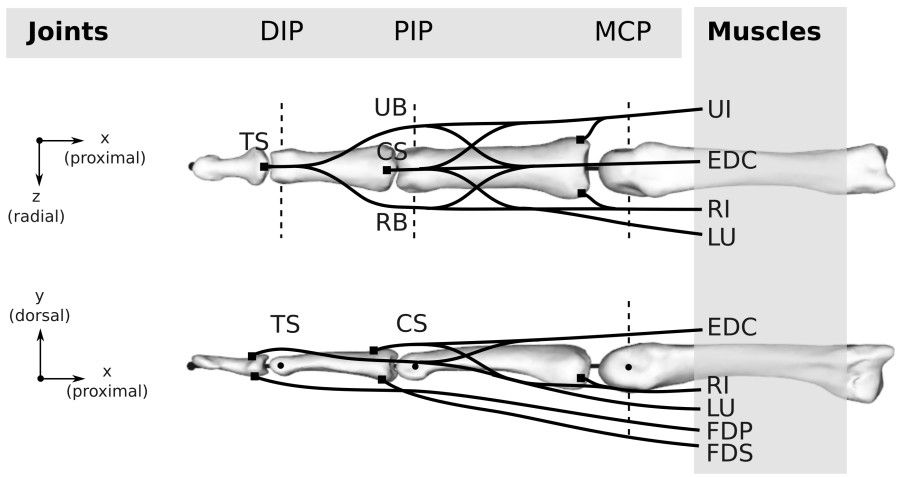

**Figure 3 Topology of both the human and bonobo model, including the kinematic description with three joints (DIP/PIP/MCP) and the six muscles (FDP/FDS/EDC/RI/UI/LU).** Dashed lines indicate the rotation axes of individual degrees of freedom of each joint. Black lines schematically indicate the tendons including the extensor mechanism, which was simplified to four tendon segments at the DIP and PIP joint, namely the terminal slip (TS), radial band (RB), ulnar band (UB), and central slip (CS). For the remaining abbreviations, the reader is referred to the main text.

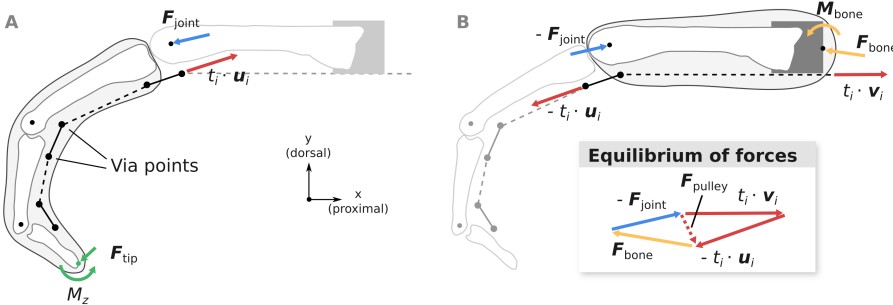

**Figure 4 Overview of the computation of joint load $F_{\text{joint}}$ (A) and the net metacarpal bone loading $F_{\text{bone}}$ (B).** Pulley forces $F_{\text{pulley}}$ are added as a dashed vector in the graphical depiction of the equilibrium conditions to highlight the differences between $F_{\text{joint}}$ and $F_{\text{bone}}$. $u_i$ and $v_i$ are unit vectors governing the tendon force directions and $t_i$ is the respective muscle tension.

distal via points were assumed to be constant in the coordinate systems of the proximal and distal bone of the articulation, respectively. Note that the joint posture still affects the line segment connecting the proximal and distal via point, leading to posture-specific moment arms (see also Fig. 1 of Article S2).

### Computation of fingertip forces

Following *Valero-Cuevas, Zajac & Burgar (1998)*, static fingertip forces and moments $F_{\text{tip}}^* = \left[ F_{\text{tip}}^{\mathrm{T}}, M_z \right]^{\mathrm{T}}$ (see Fig. 4) were computed from the tendon tensions $t =$

$[t_{\text{RI}}, t_{\text{LU}}, t_{\text{UI}}, t_{\text{FDP}}, t_{\text{FDS}}, t_{\text{EDC}}]^{\text{T}}$ using the following linear relation:

$$F_{\text{tip}}^* = -J^{-\text{T}} T t \qquad (2)$$

where $J^{-\text{T}}$ is the $4 \times 4$ inverse transpose Jacobian matrix which converts joint torques into fingertip forces and torques and $T$ is the $4 \times 6$ force transmission matrix which contains the effective moment arms of each muscle at each degree of freedom (*Lee et al., 2008*). Effective moment arms are corrected for the fraction of force transmitted to a certain part of the extensor mechanism. For instance, if only 50% of the muscle force is transmitted to a specific part of the extensor mechanism (e.g., the radial band) due to a tendon bifurcation, the respective moment arm is lowered by 50% accordingly. The moment arms of each tendon segment were computed using the generalized force method (*Sherman, Seth & Delp, 2013*) and the assumption of bowstringing between via point coordinates. Moment arms of tendon segment that would naturally wrap around the bone in a specific posture (e.g., the terminal slip in flexion, or flexor tendons in hyperextension) were computed using Landsmeer's model 1 (*Chao et al., 1989*), i.e., the moment arms were assumed to be constant for this tendon segment. This assumption leads to similar results as using wrapping geometries (Article S2). All moment arm computations were verified using the musculoskeletal modelling software OpenSim (*Delp et al., 2007*) (Article S2).

### Computation of metacarpal bone forces and MCP joint forces

Using the fingertip forces $F_{\text{tip}}$ computed using Eq. (2), the MCP joint loads $F_{\text{joint}}$ can be calculated from the static equilibrium equation:

$$F_{\text{joint}} = -\left( F_{\text{tip}} + \sum_i t_i u_i \right) \qquad (3)$$

where $t_i$ is the tension of muscle $i$ and $u_i$ is the unit vector pointing from the distal to the proximal via point of muscle $i$ (see Fig. 4A). In case tendon segments would naturally wrap around the bone in a specific posture as described in the previous paragraph, direction $u_i$ was considered constant with respect to the distal bone. Joint load computations were also verified by comparison to OpenSim (see Article S2).

Assuming that the extrinsic flexor tendons run in parallel to the long axis of the metacarpal bone as in the experiment, the net load acting on the metacarpal bone $F_{\text{bone}}$ can be computed as:

$$F_{\text{bone}} = F_{\text{joint}} - \sum_i (t_i v_i - t_i u_i) \qquad (4)$$

where $v_i$ is the unit vector parallel to the long axis of the metacarpal bone, pointing in the proximal direction (see Fig. 4B). Note that compared to $F_{\text{joint}}$, $F_{\text{bone}}$ takes into account anatomical pulley forces and forces from the tendon wrapping around the head of the metacarpal bone.

### Initial model parameters

Initial parameters of the human model were taken from literature. Normalized bone segment lengths and tendon via points were taken directly from *An et al. (1979)*. Force

distributions due to tendon bifurcations were initally set as follows: the fraction of force transmitted to the proximal phalanx and extensor mechanism was defined by the ratio of PCSA values of the muscles (e.g., 50:50 ratio for the RI muscle, see Table 1). The remaining transmission fractions were initially set to 50% at each tendon bifurcation, e.g., 50% of the RI muscle force transmitted to the extensor mechanism is transferred to the radial band, and the remaining 50% is transferred to the central slip.

The initial bonobo model parameters were obtained from a dissection study on the contralateral arm of the same bonobo cadaver used in this study (see Article S1). In brief, bone segment lengths were measured from a computed tomography (CT) scan using Blender (v2.64; Blender Foundation, Amsterdam, Netherlands) and custom Python scripts. Two via points for each tendon and each joint were then determined to obtain a description of the tendon paths consistent with that of *An et al. (1979)* for the human finger model using the following method: First, tendon paths were digitized at regular intervals relative to the closest bone using an electromagnetic motion tracking system (Patriot, Polhemus, Vt, USA) and radio-opaque markers attached to each bone. Second, tendon path points were transformed into the CT coordinate system by registering the digitized bone marker locations to those identified in the CT scan. Third, one proximal and one distal point of each tendon relative to each joint that best represented an anatomical constraint (e.g., pulley of a flexor tendon) were chosen as the final via points. Initial force transmission fractions of the extensor mechanism of the bonobo were set in analogy to the human model.

## Parameter identification

The goal of the parameter identification step was to minimize the difference between the predicted and experimentally measured fingertip forces resulting from single tendon loading in all four postures. Only via points and force transmission fractions within the extensor mechanism were included in the parameter identification since these parameters were assumed to be associated with the largest uncertainty. The model parameters $p$ were then identified by solving the following optimization problem:

$$\underset{p}{\text{minimize}} \quad \sum_{i=1}^{n}(f_i - \hat{f}_i(\boldsymbol{p}))^2 + \sum_{j=1}^{m} w_j(p_j - p_{0,j})^2 \qquad (5)$$

In the above equation, $\boldsymbol{f}$ is a one-dimensional vector containing the $n$ components of the experimentally measured fingertip forces of all postures in the x-y plane, and $\hat{\boldsymbol{f}}$ contains respective model predictions. The second term in Eq. (5) adds a penalty for large deviations of the model parameters $\boldsymbol{p}$ with respect to initial parameters $\boldsymbol{p}_0$ and should avoid obtaining unphysiological models. The $m$ model parameters contained in $\boldsymbol{p}$ comprise the x- and y-components of the tendon via points as well as the force transmission fractions of the extensor mechanism. $\boldsymbol{w}$ is a vector containing penalty weights which were manually set to 10 for all via point coordinates at the DIP and PIP joints, and to 1 at the MCP joint to qualitatively account for spatial constraints (i.e., more space is available at the MCP joint when compared to IP joints). Penalty weights of the extensor mechanism parameters were set to zero.

Since the FDS and FDP muscle parameters are independent of each other and of all other muscles attaching to the extensor mechanism, three separate optimizations were performed: (1) FDP muscle parameters ($n = 8$ fingertip force components; $m = 12$ via point parameters), (2) FDS muscle parameters ($n = 8$; $m = 8$), and (3) UI, RI, LU, EDC muscle parameters ($n = 32$; $m = 32 + 4$ via point plus extensor mechanism parameters). For the human model parameter identification, the experimental fingertip forces $f$ were averaged over all three specimens. The optimization was performed using a local optimizer (sequential least squares of SciPy (*Jones, Oliphant & Peterson, 2001*)) to obtain the best set of parameters close to the initial, physiological parameters.

The remaining overall mismatch between predicted and measured fingertip forces was quantified by the root mean square error (RMSE) of the fingertip force components:

$$\text{RMSE} = \sqrt{\frac{1}{n}\sum_{i=1}^{n}(f_i - \hat{f}_i(\boldsymbol{p}))^2} \qquad (6)$$

Additionally, the overall relative error $\text{RMSE}_{\text{rel}}$ was evaluated:

$$\text{RMSE}_{\text{rel}} = \frac{\text{RMSE}}{1/24 \cdot \sum_{k=1}^{24} \|\boldsymbol{F}_{\text{tip},k}\|} \qquad (7)$$

where $\boldsymbol{F}_{\text{tip},k}$ are the 24 fingertip force vectors of all six muscles and in all four postures. The mean magnitude of all fingertip forces was chosen as a reference value since it represents a PCSA-weighted mean of the fingertip forces generated by all muscles.

Finally, muscle-specific RMSE and $\text{RMSE}_{\text{rel}}$ were evaluated in analogy to Eqs. (6) and (7), but considering only predicted and measured fingertip forces associated with the respective muscle.

## Validation and comparison of models

The performance of the adjusted models was tested by comparing predictions of fingertip forces and net metacarpal bone forces to the experimental measurements during combined loading of the FDP and FDS tendons at two load levels. The differences were evaluated as the error of force vector magnitudes and directions in the x-y plane. Force magnitude errors were computed both in absolute values as well as relative to the experimental force magnitude.

In addition to the comparison between models and experiments, the MCP joint forces $\boldsymbol{F}_{\text{joint}}$ (see Eq. (3)) were compared qualitatively to the net metacarpal bone loads $\boldsymbol{F}_{\text{bone}}$ (see Eq. (4)) to judge the influence of tendon pulley or wrapping forces.

Finally, ratios of total muscle tension to predicted fingertip forces as well as metacarpal bone forces to fingertip force were evaluated and compared between human and bonobo models to investigate whether or not the implementation of a bonobo specific model is warranted. Bone segment lengths and average moment arms of FDP and FDS tendons at each joint were evaluated to interpret possible differences in these ratios.

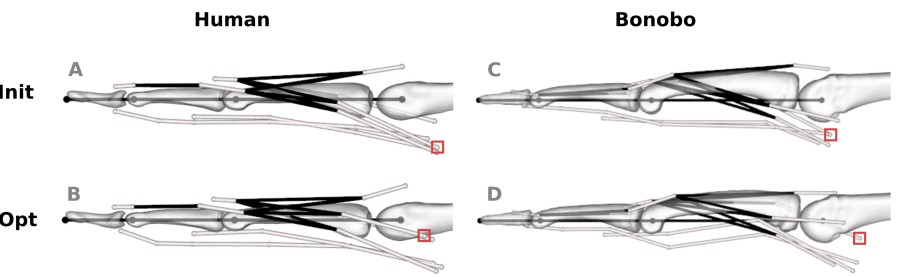

**Figure 5 Human and bonobo finger model before ("Init", A and C) and after ("Opt", B and D) optimization.** White lines represent tendon paths, as defined by via points (white spheres). Black lines represent topological tendon connections due to the extensor mechanism. The red boxes highlight selected via points of muscles that required particularly large adjustments, i.e., the radial interosseus of the human model and the ulnar interosseus of the bonobo model.

## RESULTS

### Parameter identification

Parameter identification led to physiologically plausible models both for the human and bonobo (see Fig. 5; for the full set of optimized model parameters please refer to Article S3). On average, the via points were shifted by 0.92 mm in the human model and 1.99 mm in the bonobo model, although individual points located at the MCP joint were shifted as much as 9.00 and 11.34 mm in the human and bonobo model, respectively. This shift reduced the overall RMSE (and $RMSE_{rel}$) of the fingertip forces from 0.53 N (52.10%) to 0.11 N (10.73%) in the human model and from 0.69 N (112.15%) to 0.20 N (33.24%) in the bonobo model.

The via point shifts affected both the proximo-distal (x) and volar-dorsal (y) coordinates. Average shifts along the x- and y-axis of the human model were 0.89 and 0.96 mm, respectively, and the average shifts of the bonobo model amounted to 1.93 and 2.06 mm, respectively. The via point shifts were largest at the MCP joint in both models (human: 1.57 mm; bonobo: 3.73 mm on average) and decreased towards the DIP joint (human: 0.42 mm; bonobo: 0.16 mm on average). Tendon paths that required the largest adjustments were the RI at the MCP joint of the human model (4.06 mm on average) and the UI at the MCP joint of the bonobo model (6.21 mm on average).

Comparison of measured and predicted x- and y-components of the fingertip forces from all postures and muscles of the human finger shows that the remaining error (RMSE) of fingertip forces in the optimized model was similar for all muscles (Figs. 6A–6B), ranging from 0.08 N (FDP) to 0.15 N (FDS). Relative errors ($RMSE_{rel}$) were particularly large for muscles with small PCSA such as the LU (77.84%) and low for muscle with large PCSA such as the FDP (3.55%).

Absolute RMSE values of the fingertip forces of the bonobo finger model (see Figs. 6C–6D) were again similar for all muscles, ranging from 0.14 N (LU) to 0.27 N (UI), but overall larger than in the human model. Relative errors ($RMSE_{rel}$) were again higher for muscles with smaller PCSA and ranged from 16.96% (FDP) to 85.45% (LU).

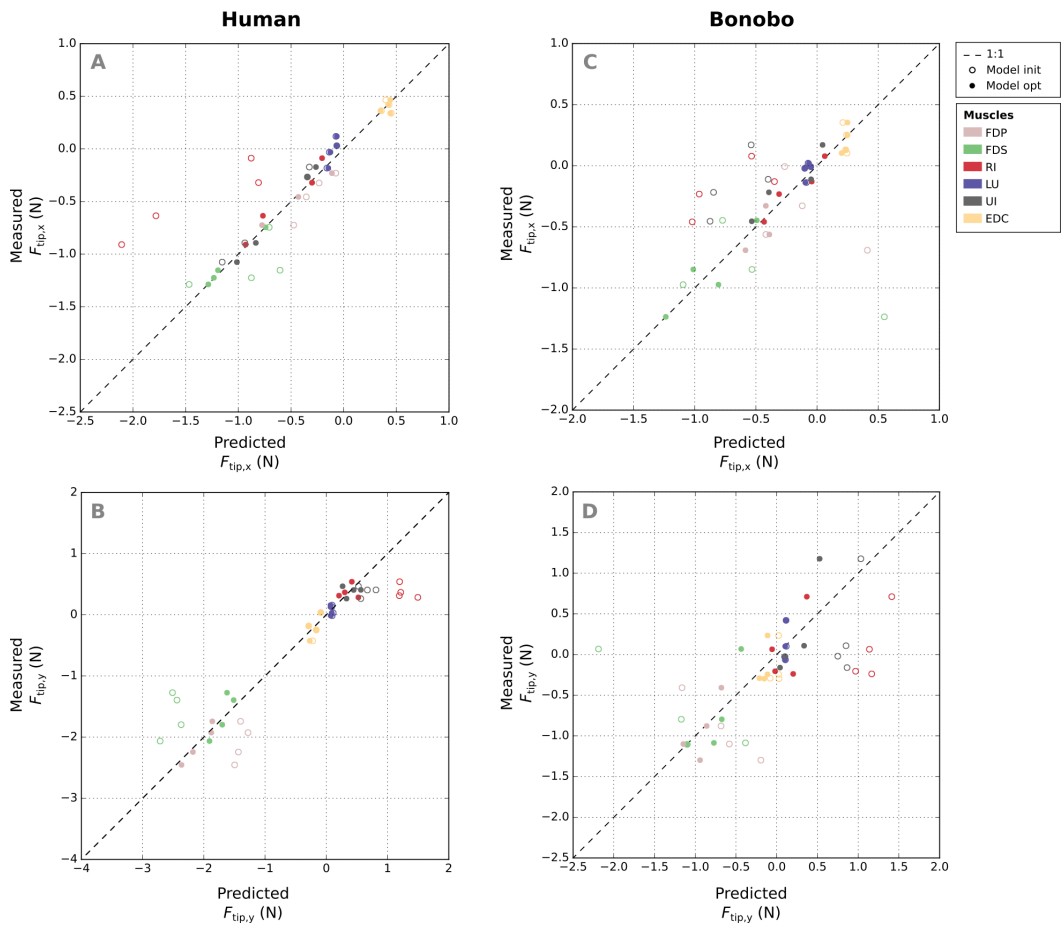

**Figure 6** **A comparison of the measured x- and y-components of the fingertip force vectors (A and C/B and D) to the human and bonobo model predictions for each muscle, both before ("init") and after parameter optimization ("opt").** Each muscle is represented by four points as fingertip force vectors were measured and predicted in four postures. EDC: extensor digitorum communis; FDP: flexor digitorum profundus; FDS: flexor digitorum superficialis; RI: radial interosseus; UI: ulnar interosseus; LU: lumbrical.

## Validation and comparison of models

Good agreement between experimental results and predictions was observed in the combined extrinsic flexor tendon loading scenario at two load levels (Fig. 7). Specifically, average directional and magnitude errors of the fingertip force vectors (human/bonobo) were 3.10°/5.76° and 0.25 N (11.03%)/0.2 N (11.70%).

Similar to fingertip forces, net metacarpal bone loads resulting from combined tendon loading were in good agreement with experimental results for both the human and bonobo finger model (Fig. 8), with average errors (human/bonobo) of 3.32°/0.57° and 0.16 N (2.34%) / 0.26 N (4.10%). Interestingly, the direction of the net metacarpal bone force vector ($F_{bone}$) showed low variability with respect to posture and was negatively correlated with MCP joint angles, i.e., higher flexion at the MCP joint resulted in more palmarly oriented net force on the metacarpal bone (see Fig. 8). In contrast, the directions of the actual joint loads $F_{joint}$ (i.e., the bone loads without tendon wrapping/pulley forces) was

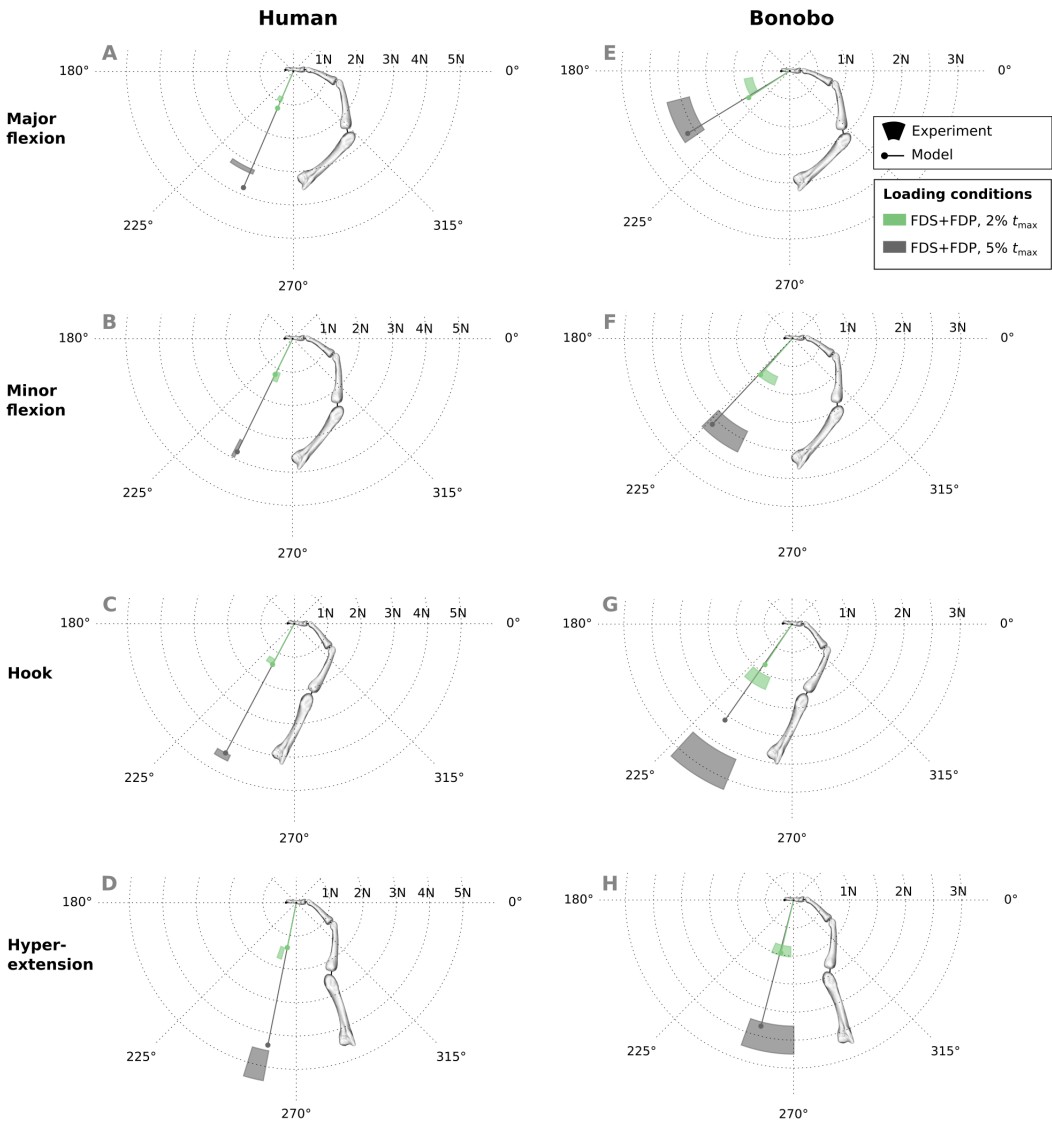

**Figure 7** **A comparison of measured and predicted fingertip force vectors engendered by combined loading of the FDP and FDS muscles at two load levels in four postures.** In the human data plots (A–D), the coloured areas represent the experimental mean ± 1 standard deviation. In the bonobo data plots (E–H), the coloured areas represent the measurement ± 10% of the magnitude and ± 10°. FDP: flexor digitorum profundus; FDS: flexor digitorum superficialis.

positively correlated with MCP joint angle and were more variable with respect to finger postures when compared to the bone loads.

Finally, the ratios of tendon load to fingertip force, as well as bone load magnitude to fingertip force were compared between the optimized bonobo and human finger model in the combined tendon loading scenario. The average tendon load to fingertip force ratio was approximately 42% higher in the bonobo (mean: 5.36; range: 5.06 to 5.66) when compared to the human (mean: 3.78; range: 3.54 to 4.10). The average ratio of bone load magnitudes to fingertip forces were approximately 55% higher in the bonobo (mean: 4.44; range: 4.19

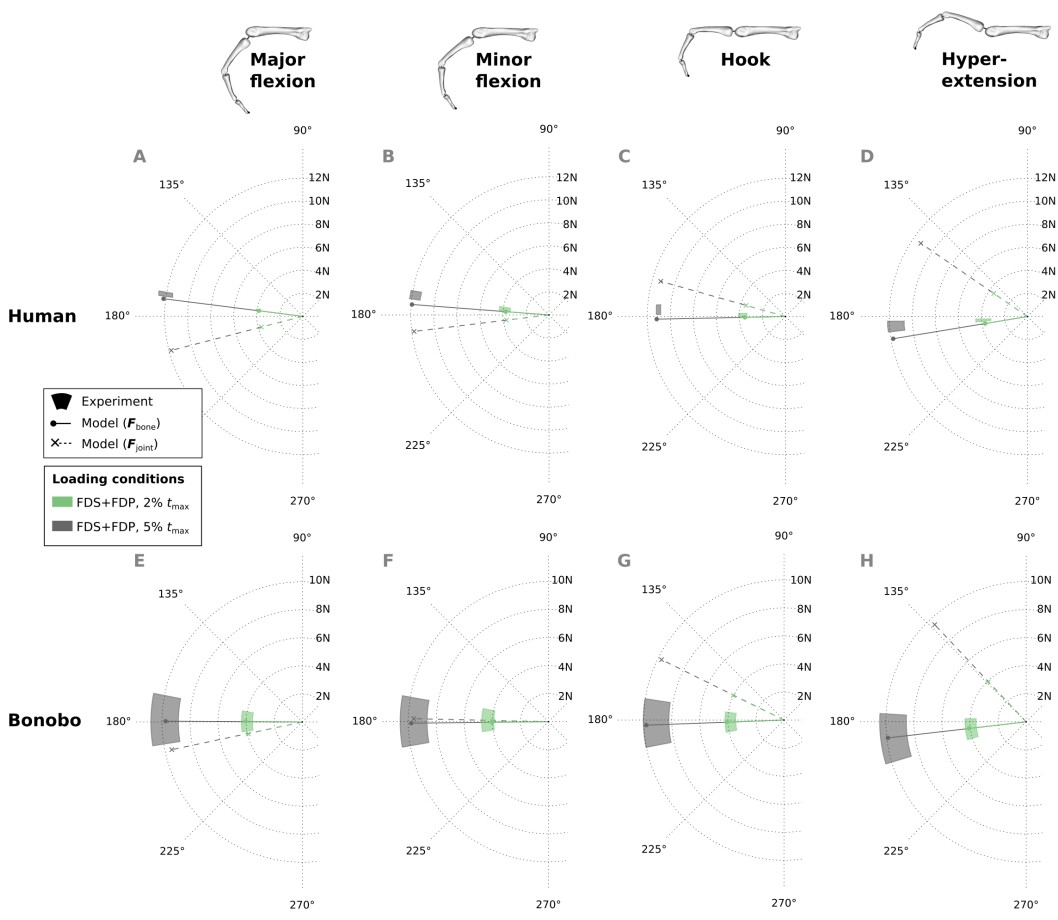

**Figure 8** Net metacarpal bone force vectors ($F_{bone}$) predicted for combined loading of the FDP and FDS muscles at two load levels in four postures (solid lines) compared to experimental measurements (coloured areas). In the human data plots (A–D), the coloured areas represent the experimental mean ± 1 standard deviation. In the bonobo data plots (E–H), the coloured areas represent the measurement ± 10% of the magnitude and ± 10°. Additionally, the MCP joint load vectors ($F_{joint}$) are plotted with dashed lines for comparison. FDP: flexor digitorum profundus; FDS: flexor digitorum superficialis.

to 4.68) when compared to the human model (mean: 2.87; range: 2.55 to 3.15). This result is consistent with the observation of larger bone segment lengths, but not necessarily larger muscle moment arms in the bonobo model (see Table 3).

## DISCUSSION

The goals of this study were (1) to identify the parameters of both a human and bonobo finger model that minimize the error of predicted fingertip forces and (2) to compare the adjusted model predictions to experimental data in different load cases for validation. The parameter identification showed that even minor parameter changes led to a substantial reduction in the predictive error, although relative errors associated with intrinsic muscles remained comparatively large. The adjusted model predictions of fingertip forces and net

**Table 3 Comparison of the human and bonobo musculoskeletal model in terms of bone segment lengths and average moment arms of flexor digitorum profundus (FDP) and flexor digitorum superficialis (FDS).** Bone segment lengths were taken from *An et al. (1979)* for the human model and own dissection data for the bonobo specimen. Moment arms are presented for each joint, but averaged over all four postures. For a full set of parameters, the reader is referred to Article S3.

| Species | Bone segment length (mm) | | | Average flexion/extension moment arm (mm) | | | | |
|---|---|---|---|---|---|---|---|---|
| | DP | MP | PP | FDP DIP | PIP | MCP | FDS PIP | MCP |
| Human | 19.0 | 28.8 | 47.0 | 4.3 | 11.1 | 12.1 | 7.3 | 12.8 |
| Bonobo | 20.4 | 38.1 | 57.8 | 3.9 | 10.4 | 12.9 | 6.6 | 13.3 |

Notes.
DP, distal phalanx; MP, middle phalanx; PP, proximal phalanx; DIP, distal interphalangeal; PIP, proximal interphalangeal; MCP, metacarpophalangeal.

metacarpal bone loads during combined loading of extrinsic flexor muscles were in good agreement with experimental measurements, leading to average errors of force direction and magnitude below 6° and 12%, respectively.

To the best of the authors' knowledge, this is the first study to identify optimal musculoskeletal finger model parameters using forces measured in vitro. Previous studies have already shown that the accuracy of moment arms of finger models can be considerably improved by adjusting via point locations (*Lee et al., 2014*) or by adding optimally positioned tendon wrapping geometries (*Kociolek & Keir, 2011*). *Qiu & Kamper (2014)* compared predicted to experimentally measured fingertip forces but needed to manually adapt proximal tendon via point locations at the MCP joint. In this study, it could be shown that a simple local optimization procedure dramatically reduces the predictive error while keeping model parameter changes to a minimum and thereby maintaining physiologically reasonable tendon paths. The large influence of even minor model parameter adjustments further highlights the parameter sensitivity of the models and warrants a careful validation procedure.

The fingertip forces resulting from intrinsic muscle (RI, UI, LU) loading generally led to larger relative errors when compared to extrinsic muscles. These errors might be caused by model simplifications but also by limitations of the experimental setup. Although previous studies used a similar experimental design and specimen preparation procedure (*An et al., 1983*), it was discovered that the removal of soft tissue at the metacarpal level influenced the intrinsic muscle tendon path to a larger extent than expected, leading to excessive bowstringing. This might also explain why particularly large adjustments of the interossei muscle via points were suggested by the model optimization. Experimental setups that keep more of the soft tissue intact were previously presented (*Qiu & Kamper, 2014*; *Valero-Cuevas, Towles & Hentz, 2000*), but applying load to intrinsic muscles remains challenging and was still not perfectly physiological in these studies. For instance, *Valero-Cuevas, Towles & Hentz (2000)* applied the load of the dorsal interosseus muscle via nylon chords attached to a screw placed at the base of the proximal phalanx. Moreover, direct measurement of metacarpal bone loads is further complicated with these experimental setups. In contrast to the intrinsic muscles, the force transmission of extrinsic flexors, which are particularly important for forceful grasping (*Long et al., 1970*; *Sancho-Bru et al.,*

*2003*; *Goislard De Monsabert et al., 2012*), could be predicted with lower relative errors. The fingertip forces and net metacarpal bone loads from the combined tendon loading regime in particular highlighted the models' good predictive abilities, with average errors of directions and magnitudes below 6° and 12%, respectively. These values are comparable to the validation results of *Qiu & Kamper (2014)*, who reported average errors of fingertip force direction and magnitude beyond one standard deviation ranging from 0 to 1.7° and from 0 to 10% for their model when compared to in vitro measurements.

The difference between the human and bonobo model was quantified by the ratios of muscle force to fingertip forces as well as net metacarpal bone load magnitudes to fingertip forces; values that are often used to quantify the efficiency of force transmission. In the literature, ratios of extrinsic flexor (FDP/FDS) muscle force to fingertip force were reported to be highly variable and posture dependent, ranging from 0.71 to 7.92 (*Dennerlein et al., 1998*; *Schuind et al., 1992*; *Kursa et al., 2005*). Although both the human and bonobo model ratios fall within this range (average of 3.78 and 5.36, respectively) and the sample size used in this study is too small to draw direct conclusions, the larger muscular effort to counteract external load in the bonobo finger can be interpreted in terms of anatomical differences. Specifically, bonobo hand bones are longer but not necessarily larger at the epiphyses when compared to humans (*Susman, 1979*), which leads to large lever arms for externally applied loads relative to the moment arms of the muscles. This was also confirmed by the comparison between the bonobo and human model in terms of total bone segment length (22% longer) and average extrinsic flexor moment arms (4–7% larger) in this study (see Table 3). Such differences cannot be captured with mere isotropic model scaling and justify the use of a species-specific set of model parameters.

Another interesting observation in this study was the direction of net metacarpal bone loading ($F_{bone}$) when compared to MCP joint loading ($F_{joint}$). The net metacarpal bone force direction varied little with posture, was mainly aligned with the long axis of the metacarpal bone, and was even slightly negatively correlated with MCP joint angle. This is in contrast to the MCP joint load directions predicted by the models presented here as well as in other studies (*Weightman & Amis, 1982*), which showed large variability and a positive correlation with the MCP joint angle. These results indicate that pulley forces play a larger role in the metacarpal bone loading than initially expected. Although individual studies claimed that modelling the tendon-pulley interaction is important to obtain realistic dynamic finger movements (*Lee & Kamper, 2009*), their effect on metacarpal bone loading has not been investigated thus far and is surprising in its magnitude. This finding has important implications for the predicted differences between species and between varied locomotor/manipulative hand postures, particularly when reconstructing hand use in fossil taxa.

Several limitations of this study remain to be mentioned. An obvious and substantial limitation is the low sample size, which was mainly due to the rarity of fresh frozen non-human ape cadavers. Still, the idea of using a simple local optimization approach to improve the accuracy of the model could be tested and general biomechanical differences of the human and bonobo finger could be investigated. Another limitation is the coarse approximation of physiological intrinsic muscle/tendon paths due to dissection of soft

tissues at the metacarpal level in the experiments. Other experimental designs (*Qiu & Kamper, 2014*; *Valero-Cuevas, Towles & Hentz, 2000*) may have kept more of the soft tissue intact but would have complicated intrinsic muscle loading and the measurement of net metacarpal bone forces. Also, the parameter identification was limited to forces in the sagittal (x–y) plane and included only parameters of the tendon paths and extensor mechanism. Other parameters, such as location and orientation of joint axes might also influence the force transmission (*Valero-Cuevas, Johanson & Towles, 2003*) and their inclusion might help to further improve the accuracy of the predictions. Furthermore, the musculoskeletal models used in this study are highly simplified both in terms of tendon path and kinematic representation. The results presented here showed that the relation between fingertip forces or net metacarpal bone loading and muscle forces could still be captured with reasonable accuracy, but further output parameters of the models need to be interpreted with caution. Also, the models were developed to investigate internal loading during static postures such as grasping objects or substrates. Answering research questions related to finger dynamics, e.g., the contribution of finger muscles to propulsion during knuckle-walking, would require more detailed models and additional validation experiments. Finally, it should be mentioned that the models currently allow computing resultant forces acting on the metacarpal bone but not stresses. Future studies should enhance the models to include predictions of stresses at the joints and entheses, which are more directly related to bone morphology.

Despite their limitations, future studies may use the musculoskeletal models to compute and compare metacarpal bone loading during activities relevant to interpreting both extant and extinct ape bone morphology. For instance, the models might help to explain why differences of bone morphology between primate species with different locomotor modes are evident but more subtle than intuitively expected (*Tsegai et al., 2013*; *Synek et al., 2018*). Making the models and parameters openly accessible should, we hope, reduce the challenges of conducting such comparative studies and inspire further research. Ultimately, the use of the musculoskeletal models and gaining a more physiologically realistic knowledge of bone loading will support a more robust reconstruction of habitual hand use from fossil bones.

## CONCLUSIONS

This study presents the first attempt to implement both a human and bonobo musculoskeletal finger model, and to optimize the models using fingertip forces measured in vitro. Although experiments and models could be further improved, good agreement between predicted and measured fingertip forces as well as net metacarpal bone loads were found upon extrinsic flexor tendon loading. Since extrinsic flexor muscles are most relevant for forceful grasping, these results suggest that the models are likely accurate enough for comparisons of joint loads engendered by human and non-human great ape activities where differences are expected to be large (e.g., tool use and suspension). Albeit compromised by sample size, the observed differences between the human and bonobo model were in line with general biomechanical considerations and indicate that the use of a species-specific set of parameters is warranted in comparative studies.

## ACKNOWLEDGEMENTS

The bonobo specimen was obtained via the Bonobo Morphology Initiative 2016 at the University of Antwerp, organized by the Centre for Research and Conservation, Royal Zoological Society Antwerp (KMDA/RZSA). Particularly Jeroen Stevens, and Zjef Pereboom, are to be acknowledged for their organisation and invitation to this initiative. We also thank the medical students who helped prepare the cadaveric specimens at the Jan Palfijn Anatomy Lab (University of Leuven).

### Funding

This work was supported by the European Research Council Starting Grant #336301. The funders had no role in study design, data collection and analysis, decision to publish, or preparation of the manuscript.

### Grant Disclosures

The following grant information was disclosed by the authors:
European Research Council Starting Grant: #336301.

### Competing Interests

The authors declare there are no competing interests.

### Author Contributions

- Alexander Synek conceived and designed the experiments, performed the experiments, analyzed the data, contributed reagents/materials/analysis tools, prepared figures and/or tables, authored or reviewed drafts of the paper, approved the final draft.
- Szu-Ching Lu conceived and designed the experiments, performed the experiments, analyzed the data, contributed reagents/materials/analysis tools, authored or reviewed drafts of the paper, approved the final draft.
- Evie E. Vereecke conceived and designed the experiments, performed the experiments, contributed reagents/materials/analysis tools, authored or reviewed drafts of the paper, approved the final draft.
- Sandra Nauwelaerts contributed reagents/materials/analysis tools, authored or reviewed drafts of the paper, approved the final draft.
- Tracy L. Kivell and Dieter H. Pahr conceived and designed the experiments, contributed reagents/materials/analysis tools, authored or reviewed drafts of the paper, approved the final draft.

### Human Ethics

The following information was supplied relating to ethical approvals (i.e., approving body and any reference numbers):

The human cadaveric hands were obtained through the Human Body Donation Programme from the Medical Faculty of the University of Leuven, Belgium. Specimens

obtained via the Human Body Donation Programme can be used for the advancement of medical education and research.

## Animal Ethics

The following information was supplied relating to ethical approvals (i.e., approving body and any reference numbers):

The bonobo sample was made available by the Antwerp Zoo by Centre for Research and Conservation, Royal Zoological Society Antwerp (KMDA/RZSA) as part of the Bonobo Morphology Initiative 2016.

## Data Availability

The experimental data is available in Table S1. The data includes all fingertip and metacarpal bone force of all specimens.

An extensive description of the bonobo dissection is available in Article S1. It includes additional details on the methods used to reconstruct the initial bonobo model parameters.

The verification of the custom Python model computations are available in Article S2. The verification was done using a simplified model and the software OpenSim.

The final model parameters of both the human and bonobo finger model are available in Article S3. This includes bone segment lengths, tendon via point locations, extensor mechanism force transmission fractions, and physiological cross sectional areas of all relevant muscles.

The Python code of the model, input data extracted from the experimental dataset, and the Python code generating the results presented in the main article are available in Dataset S1.

## Supplemental Information

Supplemental information for this article can be found online at http://dx.doi.org/10.7717/peerj.7470#supplemental-information.

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
