# Peer review of "Musculoskeletal models of a human and bonobo finger: parameter identification and comparison to in vitro experiments"

_PeerJ, doi:10.7717/peerj.7470_

## Round 0.1 · original submission · Minor Revisions

Your manuscript has had 2 very constructive reviews that raise helpful points for revisions. Importantly, both reviewers agree that the limited sample sizes are justified for the purposes of this study. The recommended revisions seem very reasonable. Please detail your responses to all points in your Rebuttal included with your revised MS. Thank you for submitting this interesting study!

Reviewer 1 ·

Basic reporting

This is an appropriately conducted and well described study, in which computational models of the human and bonobo finger are developed and validated against measured fingertip and metacarpal bone forces. The overall structure of the paper is well organized and appropriately incorporates and refers to existing work describing the anatomy and biomechanical behavior of the human and non-human primate hand. Raw data is shared and clear. Supplementary materials, including additional model verification in other model platforms, increase the confidence in the model results.

Minor comments:

Abstract: Here and generally throughout the use of metacarpal bone loading should be specified to indicate what exactly you are measuring/predicting and at what point along the bone. In the abstract in particular it is not clear that you mean force loading in the midsubstance of the bone as opposed to reaction force at the joint or stress state.
Line 92: change “…by implementing a human and bonobo musculoskeletal finger model” to “by implementing human and bonobo musculoskeletal finger models” to reflect that there are separate models.

Experimental design

This study explores whether an optimization approach can be used to develop a physiologically reasonable finger model that accurately predicts fingertip forces, and whether there are meaningful differences between the human and bonobo models to warrant the use of species-specific representation. The modeling uses appropriate established approaches and new experimental data to inform the simulations.
Experimental approaches are also rigorous and ethical use of human and non-human primate is documented. Although the sample size is quite small, this limitation is appropriately discussed and does not necessarily impact some of the conclusions regarding model sensitivity.

Validity of the findings

The raw data have been provided for the fingertip force experiment as supplementary data, and the new PCSA data for the bonobo specimen is included in the main document. The findings are supported by the data presented.

Line 295: Could you please comment in the discussion on which individual points (which muscles?) were shifted to the extreme values and if there is any significance to those with respect to which muscles are most affected by the optimization and the experiments themselves? Some of these issues are discussed with respect to the influence on the force prediction, but not the details of the optimization.

Additional comments

This an overall excellent study and manuscript. The primary limitation is the low specimen number, but most of the conclusions with regard to the sensitivity of the model predictions do not necessarily require a large sample size and the authors acknowledge this limitation appropriately.


Some additional attention should be paid in the discussion to the issue of generalizability of the models (as validated for isometric fingertip force) to tasks of interest such as those use for motivation in the introduction (such as knuckle walking). How generalizable is this model and what augmentations would be necessary to apply them for the types of applications with which you motivate the study?

·

Basic reporting

No concerns. The paper is very well written, logically organized and easy to follow, in clear, succinct and professional English. Background and rationale are well defined and supported by references to published literature.

Experimental design

No concerns. The research is appropriately justified and placed in the context of gaps in the literature on the topic. Aims are clearly defined. The experimental design, data collection and analysis are, to the best of my knowledge, without error or omissions, and appear to have been collected with utmost care. Methods are described in detail.

Validity of the findings

No major concerns. Based on the information provided, the findings appear to be valid. Sample sizes are too low to determine statistical significance of any noted differences between the species (e.g., ratio of tendon loads to fingertip forces), but as this is a model-testing paper, this is appropriate. Discussion is succinct, limitations are recognized and addressed, and speculation is kept to a minimum (though see general comments).

Additional comments

Thank you for the opportunity to review this interesting manuscript. The authors have clearly put a lot of effort into the design of the models, their validation and parameter optimization, and in the preparation of the manuscript. This paper will serve as a nice proof-of-principle that fingers can be modeled relatively accurately, but more importantly that they can and should be adapted to the species in question in biological anthropology, e.g., when inferring behavior in fossil taxa. I have only a few comments to suggest, relating to clarifications and expanding a little on the relevance to biological anthropology. Here they are, roughly in order with the text (by line):

L101 (and L417): Perhaps a sentence or two on the contexts in which the use of other species’ hand models are “warranted” would help to situate the study in the broader literature, and to justify why they would be necessary for comparative studies (e.g., because one-size-fits-all leads to potentially important underestimates of loads/stresses, etc).

L119: Body mass data for the samples would be helpful to gauge whether there are any differences in tendon morphology, architecture or PCSA.

L144-148: Could the authors explain why the loading conditions are at such a small fraction of the inferred tmax? Is it a constraint of the experimental setup (e.g., sutures rupture at higher loads, load cells have max range) or a biologically motivated decision?

L152-153: It might be worth explaining why the “unloaded” forces at the fingertip and MC are not zero, presumably because of some minor contact with the load cell or PP, respectively?

Table 2: It would be helpful to add a note to the legend here explaining that the ulnar interosseous has no separate portion that inserts onto the proximal phalanx in humans, unless there is another reason for the 0.0 values for this muscle?

L183: Could the authors clarify that the “fixed” aspect here refers to the fact that the via points cannot cross the joints in question, to contrast to the fact that they can shift proximo-distally within a bone?

L284: it would be helpful if the authors could define the direction of the shifts in broad terms, i.e., whether they are primarily shifts in the dorsoventral plane, or proximo-distally along the bone (or a combination of the two).

Figure 5: related to the above, it would be helpful to highlight a few of the via points that have shifted significantly in this figure, with arrows or boxes, especially e.g., for the via points proximal to the MCP.
Figure 6: Could the authors define a negative force in the context of these graphs? Because 0 is straddled by some values, it could be interpreted as moving from compression to tension, for example. Also, to me it seems more logical to plot measured on the y-axis, but this is a minor point.

Figure 7: Could the authors explain in the text in an appropriate location why the ‘hook’ modeling loading conditions for bonobos are so far off the experimental values? All others show good agreement except for this one.

L319 (and L403): It would be interesting if the authors could include information on stresses rather than loads, since they have CT data for at least the bonobo, they might be able to provide estimates of the joint surface areas at the MCP over which these loads are dissipated. Loads are one thing, but the stresses (loads/areas) would be more relevant to interpreting/reconstructing extinct hominin behavior as it is most likely these that would be reflected in, e.g., thicker cortices or more trabecular bone in the epiphysis. If the authors don’t have the data to estimate MCP area, then at least some acknowledgment that stresses are relevant too should be made at L319 or in the discussion at L403.

Minor grammatical/wording issues:

Abstract: In Conclusions, change “first attempt of implementing” to “first attempt to implement”

L56: Suggest replacing “investigate” with “understand the etiology” or “diagnose”

L80: Technically, there are no “intrinsic finger muscles”, which would be muscles originating and inserting in the finger. Change to “intrinsic hand muscles”

L329: change “ author’s ” to “ authors’ “

L366: “wider” is an ambiguous term in the context of epiphyseal size, and could be interpreted as the mediolateral dimension, which is not as relevant here as the dorsoventral plane, which I assume the authors are referring to, and is the proper plane in which to measure the moment arms of, e.g., the digital flexors. On that note, this statement conflicts somewhat with the data in Table 3 that suggests the moment arms of the FDs at the MCP joint are substantially greater in bonobos than in humans. This is another reason why having an estimate of MCP joint surface area could be relevant to identifying species differences in bone loading.

---

## Round 0.2 · accepted · Accept

Well done-- both reviewers are now fully satisfied with the revised MS and so am I. This was a smooth peer review process in my view; I hope you agree and found PeerJ to be a good venue for your excellent science.

Reviewer 1 ·

Basic reporting

No comment

Experimental design

No comment

Validity of the findings

No comment

Additional comments

The authors have appropriately addressed the concerns described in the prior review. I have no further concerns.

·

Basic reporting

No further comments, the authors have done an excellent job addressing reviewer concerns and suggestions.

Experimental design

No concerns

Validity of the findings

No concerns

Additional comments

Thank you for the thorough revision of the manuscript, I believe it is much improved and I have no further concerns. I recommend publication, and congratulations to the authors on a great study.